# Microglia in Prion Diseases: Angels or Demons?

**DOI:** 10.3390/ijms21207765

**Published:** 2020-10-20

**Authors:** Caterina Peggion, Roberto Stella, Paolo Lorenzon, Enzo Spisni, Alessandro Bertoli, Maria Lina Massimino

**Affiliations:** 1Department of Biomedical Sciences, University of Padova, 35131 Padova, Italy; alessandro.bertoli@unipd.it; 2Food Safety Division, Department of Chemistry, Istituto Zooprofilattico Sperimentale delle Venezie, 35020 Legnaro (PD), Italy; rstella@izsvenezie.it; 3Department of Integrative Medical Biology (IMB), Umeå University, 901 87 Umeå, Sweden; paolo.lorenzon@umu.se; 4Department of Biological, Geological, and Environmental Sciences, University of Bologna, 40126 Bologna, Italy; enzo.spisni@unibo.it; 5Padova Neuroscience Center, University of Padova, 35131 Padova, Italy; 6CNR Neuroscience Institute, Department of Biomedical Science, University of Padova, 35131 Padova, Italy

**Keywords:** prion protein, neuroinflammation, microglia, prion diseases, cytokines

## Abstract

Prion diseases are rare transmissible neurodegenerative disorders caused by the accumulation of a misfolded isoform (PrP^Sc^) of the cellular prion protein (PrP^C^) in the central nervous system (CNS). Neuropathological hallmarks of prion diseases are neuronal loss, astrogliosis, and enhanced microglial proliferation and activation. As immune cells of the CNS, microglia participate both in the maintenance of the normal brain physiology and in driving the neuroinflammatory response to acute or chronic (e.g., neurodegenerative disorders) insults. Microglia involvement in prion diseases, however, is far from being clearly understood. During this review, we summarize and discuss controversial findings, both in patient and animal models, suggesting a neuroprotective role of microglia in prion disease pathogenesis and progression, or—conversely—a microglia-mediated exacerbation of neurotoxicity in later stages of disease. We also will consider the active participation of PrP^C^ in microglial functions, by discussing previous reports, but also by presenting unpublished results that support a role for PrP^C^ in cytokine secretion by activated primary microglia.

## 1. Introduction—The Prion Protein, Prions and Prion Diseases

The cellular prion protein, PrP^C^, is a glycosylphosphatidylinositol (GPI)-anchored protein, residing in the outer leaflet of the plasma membrane (PM), expressed in several cell types and particularly abundant in the central nervous system (CNS) and immune cells [1].

PrP^C^ is primarily renowned for being the precursor of prions, the proteinaceous infectious agents lacking nucleic acids that cause invariably fatal neurodegenerative disorders named transmissible spongiform encephalopathies (TSEs) or prion diseases in humans and other mammalian species [2,3,4]. Prion diseases are rare disorders, affecting about one or two people per million per year world-wide, nevertheless they attract remarkable attention due to the unique biology of the transmissible agent. Prions form upon the conversion of PrP^C^ into an insoluble, protease-resistant conformer (PrP^Sc^) that is the major, if not unique, component of the infectious particle. Although all TSEs share the presence of PrP^Sc^ aggregate deposition, they present with a variety of etiological and neuropathological traits, affecting different CNS areas, and causing various clinical manifestations.

The great majority of human prion diseases arise sporadically (e.g., sporadic Creutzfeldt–Jakob disease (CJD)), but a significant percentage (~10%) is of genetic origin, due to the autosomal dominant transmission of inherited mutations in the PrP^C^-coding gene (e.g., genetic CJD, Gerstmann–Sträussler–Scheinker syndrome and fatal familial insomnia) and about 5% of all cases develop on infectious grounds (e.g., Kuru, iatrogenic CJD and a new variant form of CJD that was transmitted to humans through the consumption of meat from bovine spongiform encephalopathy-affected cattle) [5].

The classical diagnostic triad of TSEs comprises spongiform vacuolation of the cerebral grey matter, neuronal loss, and chronic neuroinflammation [6]. Regarding the latter aspect, activation and proliferation of microglia and astrocytes have been recognized as obligatory features of the disease, regardless of the TSE form (reviewed in [7,8,9]). However, the determination of the precise role of glia in prion diseases and other neurodegenerative diseases, which are commonly associated with chronic neuroinflammation, is still a matter of debate [10,11,12,13,14]. Taking this context, it is worth remarking that the cellular mechanisms leading to brain damage in TSEs are far from being understood, and it is not yet clear if they rely on a gain in the toxicity of PrP^Sc^ aggregates or a loss of function of PrP^C^ (due to its continuous conversion into PrP^Sc^), or both. Concerning the loss of function hypothesis, the understanding of PrP^C^ physiological function would be of paramount importance. While both the high evolutionary conservation and the almost ubiquitous expression of PrP^C^ would suggest fundamental roles for the protein, and in spite of decades of extensive research, a comprehensive view of PrP^C^ function, however, is still missing.

The possible involvement of PrP^C^ in microglia pathophysiology and the role of microglia in PrP^Sc^ infection and propagation during prion disease are the major topic of this review. Particularly, we will discuss previous controversial results on the role of PrP^C^ on microglial activation and cytokine release, and also provide our unpublished data.

## 2. Microglia Function in Health and Prion Diseases

### 2.1. Microglia Origin and Function in the Healthy Brain

Microglia belong to the glial system of non-neuronal cells and represent the resident immune population of the CNS. Although it has been studied for decades, the developmental origin of microglia remained debated for a long time [15]. Recent studies confirmed the prediction of the founder of the microglia field, Pio del Rio–Hortega [16], indicating that microglial cells are derived from c-Kit^+^ erythromyeloid precursors in the yolk sac that seed the CNS rudiment from the cephalic mesenchyme very early during embryogenesis and continuing until the blood–brain barrier is formed, following a gradual process of differentiation into highly specialized immune cells in the brain [17,18]. After such a developmental origin, self-renewal is the only source of new microglial cells in the healthy brain, which is regulated also by astrocytes and neurons through the activation of the microglial colony-stimulating factor (CSF)-1 tyrosine kinase receptor (CSF-1R) by its ligands CSF-1 and interleukin (IL)-34 [19].

Microglial cells play a key role in the maintenance of brain homeostasis from early development to adulthood, including neurodevelopment, synaptic pruning, neuronal circuit maturation [20,21], and the impairment of microglial functions which can lead to severe pathological outcomes.

Under healthy conditions, microglial cells are characterized by a small cell soma and numerous branching processes involved in the clearance of metabolic products and apoptotic cells or cell debris [22]. Key surface receptors (i.e., cluster of differentiation (CD) 45, CD14, and CD11b/CD18 (Macrophage-1 antigen)) maintain microglial cells in a resting but highly dynamic state [23] that is favored by the interaction with neurons, for example through the formation of a molecular complex between the neuronal transmembrane glycoprotein CD200 and its receptor CD200R present in the plasma membrane of microglial cells [24], or between the neuronal chemokine C-X3-C motif chemokine (CX3C) ligand 1 (CX3CL1, also named Fractalkine) and its receptor CX3CR1, expressed solely by microglial cells [25].

### 2.2. Microglia Involvement in Neuropathology

Under pathological circumstances, such as brain injury, microbial infection, or neurodegeneration, microglial activation leads to morphological changes, up-regulation of surface receptors, secretion of a multitude of cytokines, chemokines, and reactive oxygen species (ROS) [26], and the acquisition of a phagocytic phenotype [27]. Most in vitro and in vivo studies used the bacterial cell wall endotoxin lipopolysaccharide (LPS) (reviewed in [28,29,30]) or pro-inflammatory cytokines (e.g., interferon-γ (IFNγ) and tumor necrosis factor-α (TNF-α)) to achieve microglial activation [31].

Activated microglia could exist in a range of activation states that span from two opposite phenotypes, the pro-inflammatory (M1) and the anti-inflammatory (M2) phenotype, with the phenotype of activated microglia falling somewhere along this spectrum, depending on incoming signals. A precise balance between M1 and M2 is of paramount importance for the resolution of inflammation. Indeed, the M1 phase is characterized by the secretion of pro-inflammatory mediators, such as TNF-α, IL-1β, IL-6, nitric oxide (NO), ROS, chemokines (e.g., macrophage inflammatory protein 1α, monocyte chemoattractant protein-1 (MCP-1) and the IFN inducible protein 10) and neurotoxins. When such a strong pro-inflammatory response protracts over time (e.g., under chronic conditions such as neurodegenerative disorders), the sustained release of inflammatory mediators and oxidative and nitrosative stress may exacerbate neuronal damage [13,32,33], a circumstance that may well occur in prion diseases (see below, [9]). Conversely, in the M2 response, microglia exert numerous beneficial effects through the release of anti-inflammatory factors (e.g., IL-4, IL-10, transforming growth factor-β (TGF-β), insulin-like growth factor 1 and brain-derived neurotrophic factor) [34,35], protecting against brain damage.

### 2.3. Microglial Proliferation/Activation in Prion Diseases

An active involvement of microglial activation and cytokine signaling in prion diseases has been suggested by a wealth of observations reporting different microglial responses in human CJD [36,37], in bovine spongiform encephalopathies [38], in scrapie [39], in animal models of the diseases (i.e., rodents [40,41,42] or sheep [43] infected with experimental prion strains, or prion mimetics, see below) and in cultured microglial cells exposed to PrP^Sc^ [40,44].

Both in patient and in animal models of prion diseases, PrP^Sc^ aggregates have been found inside and around microglial cells, inducing activation that correlates temporally with the onset and development of clinical and molecular signs of the disease [40,45,46,47,48,49,50,51,52,53].

The huge experimental effort, however, failed to precisely define the microglial role in prion disease progression (for a comprehensive review, see [7]). It is well-accepted, on one hand, that microglial activation begins in the early stages of disease, preceding neuronal loss and spongiform neurodegeneration, which would suggest that microglial activation is a cause rather than a consequence of neuronal demise (reviewed in [9]). A wealth of evidence, on the other hand, supports the hypothesis that microglia play a neuroprotective role in prion disease-related neuronal damage [7].

One of the most exploited models for studying prion diseases is that involving the use of mice infected with different prion strains, which recapitulated most pathological features of TSEs. Using this experimental setting, the involvement of microglia in prion diseases was principally investigated in mice in which microglial activation and cytokine signalling was altered either genetically or pharmacologically [54,55,56].

Probably due to the variety of the experimental paradigms, however, such approaches provided conflicting conclusions on the role of microglia in prion disease pathogenesis, as discussed below.

Some studies showed a beneficial role of microglia activation in prion diseases. They demonstrated, for example, that Toll-like receptors (TLRs) are involved in prion-induced microglial activation, exploiting a protective function during disease pathogenesis [7] and suggesting that TLR signalling controls the progression of prion disease, as also indicated by the finding that a loss-of-function mutation in TLR4 decreased survival after prion infection [57].

Conversely, however, it also has been suggested that the chronic activation of microglia might have a detrimental effect in the defence against prions. Particularly, a key role in the sustained maintenance of activated microglia in prion disease is retained by the CSF-1R signalling pathway. Indeed, a reduction of proliferating microglia was observed in PrP^Sc^-infected mice treated with the selective inhibitor of CSF-1R, GW2580, with a consequent slowdown in neuronal damage and disease progression [56].

Furthermore, the expression of the microglial GPI-anchored protein CD14, a TLR co-receptor involved in microglia activation, significantly increases in mice infected with different prion strains, as demonstrated by large-scale transcriptomic studies [58,59]. Interestingly, PrP^Sc^-infected CD14 knock-out (KO) mice survived longer and expressed more anti-inflammatory cytokines (such as IL-10 and TGF-β [54], IL-13 [60]) and less pro-inflammatory IL-1β than PrP^Sc^-infected wild-type (WT) mice [54], suggesting a harmful role for CD14-mediated signalling in prion pathogenesis (Figure 1).

Another key issue is the involvement of astroglial cells in prion replication, and propagation to neighboring cells (i.e., neurons) and throughout the CNS, which has been proposed to occur in prion diseases [34,61] and other neurodegenerative disorders associated with prion-like proteins [62]. While the direct participation of astrocytes and neurons in PrP^Sc^ replication and spread is well recognized [43,63,64,65,66,67,68,69], the role of microglia in such processes is still debated [34]. The low basal levels of PrP^C^ in microglia may suggest that these cells unlikely support the PrP^C^-to-PrP^Sc^ transition and act as foci of prion propagation. Nonetheless, the increase of PrP^C^ levels in microglia under inflammatory conditions, such as those occurring during prion infection or—experimentally—upon microglia activation by LPS (see below), may enhance the capability of microglial cells to favor prion replication and spreading [42,70].

Taking this context, it also is worth noting that in other neurodegenerative proteinopathies, sharing with prion diseases the molecular mechanisms of propagation of aberrant protein conformers [71,72], the contribution of microglia in cell-to-cell transmission of proteinaceous neuropathogens has been proposed. As examples, this is the case for α-synuclein in Parkinson’s disease models [73], and of the tau protein in Alzheimer’s disease-related tauopathies [74,75,76].

### 2.4. Cytokines/Chemokines Signalling Alteration in Prion Diseases

Another important and widely studied issue related to microglial response in prion diseases concerns the role of cytokines and chemokines. As previously described, both pro- and anti-inflammatory cytokines are increased in the CNS in response to prion infection. These molecules are produced by activated microglia and astrocytes, since leukocytes from the periphery do not infiltrate the CNS in prion diseases [9], and exert different functions in the inflammatory response to brain injury, as described above.

Regarding whole brain or isolated microglial cells of PrP^Sc^-infected mice, an upregulation of anti-inflammatory cytokines and chemokines (principally in the early stages of prion infection) [10,41,77], or of pro-inflammatory ones [78,79,80,81,82,83,84,85,86,87] have been observed. The different results obtained in such studies may be attributable to several factors, including the used prion strains, the stage of disease, different mouse genetic backgrounds, or techniques applied for the analyte detection.

Among the pro-inflammatory cytokines, IL-1 seems to play a prominent detrimental role in prion-associated neuroinflammation, since knocking-out the IL-1 receptor 1 (IL-1R1) prolongs prion incubation times and delays disease progression in scrapie infected IL-1R1 KO [88,89].

The role of anti-inflammatory cytokines in prion diseases mainly has been investigated by means of IL-4, IL-10, and IL-13 KO mice. Among these cytokines, the only one playing a protective role in prion diseases is IL-10, as suggested by the finding that prion-inoculated IL-10 KO mice have a faster onset and progression of the disease. IL-10 absence also favors TNF-α expression, thereby promoting a sustained pro-inflammatory response [90]. However, conflicting results were obtained in IL-10 KO mice with a different genetic backgrounds [89], supporting the idea that the IL-10-neuroprotective effect is strongly dependent on the genetic context (Figure 2).

Also, numerous chemokines (e.g., MCP-1, Regulated upon Activation, Normal T cell Expressed, and Secreted (RANTES), C-X-C motif ligand (CXCL)3, CXCL9, CXCL10) and chemokine receptors (CCR) (e.g., CCR1, CCR5 and CCR3) are up-regulated in prion diseases, but their role in the pathology remains unclear since different studies provided conflicting results [7]. Signalling pathways activated by CXCL9 and CXCL10 seem to directly contribute to prion disease progression, as suggested by the finding that scrapie infected-CXCR3 KO mice (lacking the CXCL9 and CXCL10 receptor), although exhibiting exacerbated astrocytosis and accelerated accumulation of PrP^Sc^, showed reduced microglia activation and pro-inflammatory factor secretions, and survived longer compared to scrapie-infected WT mice [91]. To contrast, prion infected-CCR1 KO mice (lacking the RANTES receptor) showed a worsened prion disease course and a lower survival rate with respect to infected WT mice [92].

Considering this, animal models provided conflicting results also for the involvement of the CX3CL1/CX3CR1 signalling axis in prion neuropathology. While Grizenkova et al., [55] suggested that CX3CR1 (a receptor for the CX3CL1 chemokine) activation is neuroprotective, a subsequent study demonstrated no contribution of the receptor to disease progression [93].

### 2.5. Microglial Activation by the Prion Mimetic PrP106-126 Neurotoxic Peptide

Considering the above reported controversial results provided by animal models, it is worth discussing a simpler experimental model that has been widely employed to study the microglia contribution to prion-related neurodegenerative processes. We specifically refer to the use of the fibrillogenic and neurotoxic peptide 106-126 derived from the human prion protein sequence (PrP106-126) [94], characterized by its intrinsic capability to form fibrils in vitro [95] and to induce a PrP-dependent neuronal cell death [96] by the activation of apoptotic cell death pathways [97]. Interestingly, the PrP106-126 neurotoxic effect requires the presence of microglia [98] and was demonstrated to be independent from de novo generation of PrP^Sc^ [99].

Particularly the PrP106-126 peptide, or other longer peptides containing the PrP106-126 sequence, induced the activation and proliferation of immortalized microglial cells [100,101,102] and primary microglia cultures [103,104,105,106,107,108,109,110,111], and stimulated astroglial proliferation [112,113]. Furthermore, such fibrillogenic peptides induced the secretion of different pro-inflammatory cytokines (i.e., TNF-α, [105]; RANTES, Granulocyte-CSF, and IL-12, [100]; IL-1β, IL-6, [104,106]), and caused an increase of NO synthase and NO release that appears to be essential for the mediation of neurotoxicity [98,105], as demonstrated by the neuronal death induced in neurons co-cultured with microglia treated with the PrP106-126 peptide [106] (Figure 2).

Taken together, the above summarized findings—based on the use of PrP-derived peptides—support the contention that microglial cells contribute to prion-related neurodegenerative processes by producing pro-inflammatory cytokines and oxidative stress, which are recognized as mediators of neuronal death. Nonetheless, it is worth underlining that some doubts on the relevance and validity of results obtained with prion protein-derived peptides were forwarded, since such studies often disregarded the non-infectious nature of PrP106-126, its absence in clinical cases, or its spontaneous production in in vivo experimental settings [7].

## 3. A possible Role for PrP^C^ in Neuroinflammation

Until now, we have presented and discussed previous studies aimed at relating microglia activation and cytokine signalling in prion neuropathology in animal and cell models challenged with prions or prion surrogates. However, in line with the loss-of-function hypothesis in prion diseases, it also is plausible to consider that PrP^C^-to-PrP^Sc^ conversion and the consequent irreversible recruitment of the protein into newly formed prions may severely perturb physiologic PrP^C^ functions in microglia and/or microglia-neuron crosstalk. The most reliable and exploited model for investigating such a hypothesis is the use of PrP-KO mice.

Although no phenotypic disturbances were originally recognized in PrP-KO mice [114,115], further investigation revealed numerous subtle phenotypes in different PrP^C^-deficient mice, many of which were exacerbated upon stressful conditions [116,117]. These models suggested pleiotropic functions for PrP^C^ [1,116], among which was the involvement in the inflammatory response both in the CNS and peripheral extra-neural tissues [118,119,120,121,122,123,124]. The protective role of PrP^C^ in the inflammatory response also was suggested by the higher expression of the protein in the so-called immuno-privileged sites, such as the CNS, eyes, placenta, fetus and testicles [125]. Despite controversial data, it is conceivable that PrP^C^ is involved also in the regulation of phagocytosis, a fundamental process for both the resolution of inflammation and the immune response against pathogens [126,127,128,129].

Interestingly, PrP^C^ seems to regulate the expression and/or the secretion of pro- or anti-inflammatory cytokines after systemic LPS administration in vivo, although the involved signalling pathways and mechanisms are largely unknown [121,123,124,130]. Recently, a new scenario emerged demonstrating that PrP^C^ increases TNF-α production through the stimulation of pro-TNF-α cleavage (as demonstrated in serotoninergic neurons [131] and muscle cells [132]) by the activation of the TNF-α converting enzyme (TACE), a member of the A-Disintegrin-and-Metalloproteinase family, via its coupling to nicotinamide adenine dinucleotide phosphate (NADPH) oxidase. Alternately, by activating the same NADPH oxidase-TACE axis and by modulating TACE localization in the plasma membrane, PrP^C^ also may promote the shedding of transmembrane TNF-α receptors, thereby protecting against excessive pro-inflammatory TNF-α response [133].

Unfortunately, even such studies aimed at elucidating PrP^C^ function in microglial cells using PrP-KO models provided contradictory data. Indeed, while early studies on immortalized microglial cells suggested a key role for PrP^C^ in microglial activation and regulation of the inflammatory response [134,135], more recent data, obtained in murine primary microglial cultures stimulated with LPS, reported no difference in cell morphology, microglial marker expressions and cytokine production between cells expression, or not, PrP^C^ [136].

Viewing this puzzling context, we analyzed the influence of PrP^C^ in microglial function and cytokine production using a mouse model with a different genetic background with respect to those previously employed. Starting from cortical microglial primary cultures from 2–3 post-natal day transgenic (Tg) mice expressing (PrP-Tg) or not PrP^C^ (PrP-KO), we induced microglia activation by adding LPS to the culture medium. Interestingly, LPS-treated (for 96 h) PrP-Tg microglial cells expressed higher PrP^C^ amounts compared to non-treated cells (Figure 3A), as already observed in other cell models [130,137,138]. This result corroborated the hypothesis of a key role of PrP^C^ in the microglial activation process.

Using the same experimental setting, we evaluated the secretion of pro- (IL-1β, IFN-γ, IL-6, TNF-α) and anti-inflammatory (IL-10) cytokines at 24 h and 96 h after LPS-activation of microglia expression, or not, PrP^C^. We chose this restricted set of cytokines because they were found to be de-regulated in CJD patients and prion infected mice [7], suggesting their involvement in prion neuropathology.

Shown in Figure 3B, our results demonstrated that, in our experimental paradigm (PrP-KO and PrP-expressing mice with an almost pure FVB genetic background, [139]), the secretion of both pro- and anti-inflammatory cytokines was higher in PrP-expressing microglia after 24 h of LPS treatment, while after prolonged inflammatory stimulus, only pro-inflammatory cytokine (IL-1β, IFN-γ, IL-6, and TNF-α) release was stimulated by the presence of PrP^C^ (for experimental details, see the legend to Figure 3).

Such findings are in agreement with previous data obtained from immortalized microglia-like cell line, expressing or not PrP^C^, in which it was shown that PrP^C^ expression alters TNF-α and IL-1β [134,135] or TGF-β and IL-10 [134] secretion. However, our results are in contrast with data obtained by Pinheiro et al., [136], reporting that genetic ablation of PrP^C^ did not affect LPS-induced production of pro-inflammatory (TNF-α, IL-6, IL-1β) and anti-inflammatory (IL-10) cytokines.

We conceive that the discrepancy between our observations and the previously reported results may originate from the different genetic background of the mouse models (FVB, our experiments; B10.129Ola, Pinheiro and co-workers). It has been suggested, for example, that a polymorphism in the *SIRPA* gene, encoding the signal regulatory protein (Sirp) α, can be responsible for many phenotypes often associated to PrP^C^ ablation in different mouse genetic backgrounds [140]. To the best of our knowledge, however, the *SIRPA* genotype was never analysed in the PrP-KO FVB strain that we have employed in our studies, thus further investigation is needed to understand the possible role of Sirpα, or a Sirpα-PrP^C^ cross-talk, in microglial activation and cytokine production.

## 4. Concluding Remarks

As described in this review, microglia are likely to play a relevant role in prion neuropathology, but—in light of the conflicting data accumulated over time—it is yet to be clearly understood if microglia play as a beneficial actor in, or an enhancer of, prion-related neurodegeneration.

One of the most accredited pictures emerging from the above discussed reports envisages that—in the first stages of the disease—microglial cells act as a suppressor of neurotoxicity induced by PrP^Sc^ deposits by facilitating their removal and clearance and by secreting anti-inflammatory factors that prevent neuronal loss. However, in an advanced stage of disease, microglia would no longer be able to contrast the effects of prolonged PrP^Sc^ accumulation and neurotoxic signalling, which would rather elicit the switching of microglia to a sustained/chronic pro-inflammatory response, thus worsening brain damage.

Taking this view, it also is worth considering that the depletion of functional PrP^C^ in microglia and microglia-neuron crosstalk may exacerbate the effects of microglia misregulation, thereby contributing to disease progression. Also in this case, however, past literature and the here-presented original data offer a panel of conflicting data, thus highlighting once more the need for a deeper understanding of the physiological PrP^C^ function in different cellular environments and signalling pathways.

It is our opinion that a unifying definition of microglia role in prion disease pathogenesis is crucial for both getting major insights into the neurodegenerative processes and developing suited therapeutic strategies. Particularly, a prompt stimulation of the anti-inflammatory properties of microglia in the first stages of prion diseases may represent an optimal target to ameliorate disease progression and the quality of life of patients.

## Figures and Tables

**Figure 1 ijms-21-07765-f001:**
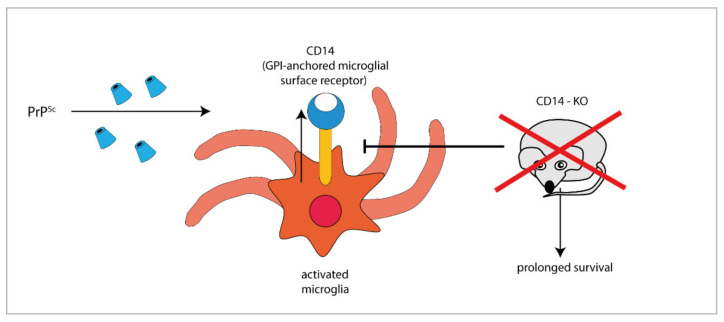
The GPI-anchored cell surface TLR co-receptor CD14 is upregulated in microglial cells during prion infection, while CD14 genetic deletion blocks (T-arrow) CD14 upregulation and prolongs survival in prion infected knock-out mice (red × symbol).

**Figure 2 ijms-21-07765-f002:**
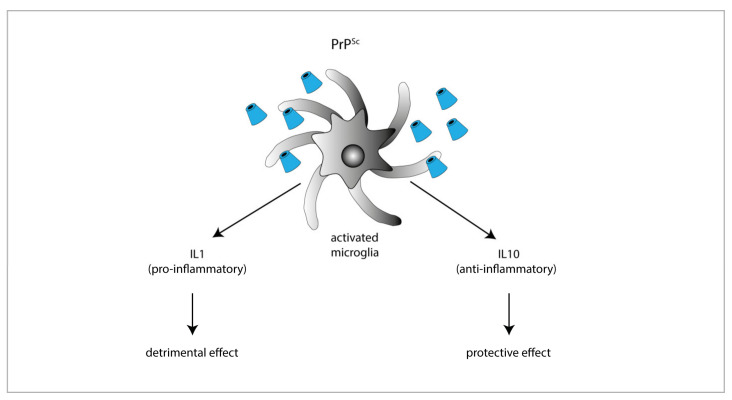
Exposure to PrP^Sc^ or prion mimetics (e.g., the fibrillogenic 106-126 peptide) activates microglial cells promoting the secretion of either pro-inflammatory IL-1 (resulting in enhanced neurodegenerative cues), or anti-inflammatory IL-10 (exerting neuroprotective effects).

**Figure 3 ijms-21-07765-f003:**
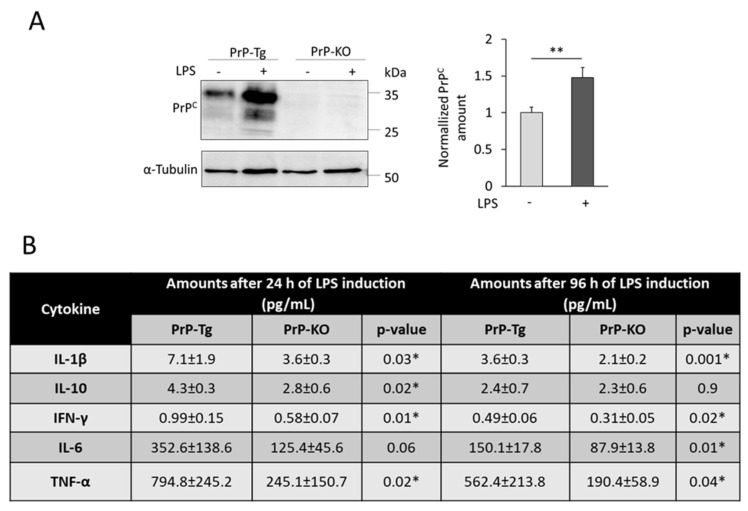
(**A**) LPS treatment increased PrP^C^ expression in primary murine microglial cells. Primary microglial cells were isolated from PrP-KO mice with an almost pure FVB genotype (strain F10), and PrP-Tg mice in which physiologic PrP^C^ expression levels were rescued over the F10 genetic background (strain Tg46) [139]. Both mouse lines were kindly provided by the MRC Prion Unit at UCL, London, UK. Cell cultures were established and maintained as described previously. [141]. All procedures were performed in compliance with European and Italian (D.L. 26/2014) laws concerning the care and use of laboratory animals and were approved by the Italian Ministry of Health, and by the Ethical Committee of the University of Padova (Authorization n. 743/2017-PR). Cells were cultured for 96 h in the absence (–) or in the presence (+) of lipopolysaccharide (LPS) (500 ng/mL), then cells were lysed, and extracted proteins were subjected to Western blot (WB), as described previously [142]. The left panel shows a representative WB, out of three biological replicates (i.e., different primary cell cultures), using a mouse monoclonal antibody to PrP^C^ (clone 8H4, Abcam). A mouse monoclonal anti-α-Tubulin antibody (clone B-5-1-2, Sigma–Aldrich) also was used to check for the loading of equal protein amounts. The right panel reports the densitometric analysis of PrP^C^ immuno-reactive bands in control (–) or LPS-treated cells (+), normalized to the corresponding signal of α-Tubulin. Data were then normalized to PrP^C^ amounts in control cells. Data show that LPS addition significantly increases PrP^C^ expression levels in PrP-Tg-derived microglial cells. As expected, no PrP^C^ reactive band was observed in PrP-KO samples, confirming the specificity of the immunosignal. Values are expressed as mean ± standard error of the mean (SEM), *n* = 3. Statistical analysis was based on unpaired two-tailed Student’s *t*-test. ** *p*-value < 0.01. (**B**) PrP^C^ regulates the production of cytokines by primary murine microglial cells upon LPS treatment. Microglial cells (as in panel A) were maintained in culture in the presence of LPS (500 ng/mL) for 24 h or 96 h. The amounts of selected cytokines (IL-1β, IL-10, IFN-γ, IL-6 and TNF-α) released in the culture medium were quantified by a customized enzyme-linked immunosorbent assay (ELISA) detection kit. The assays were performed in 96-well filter plates by Multiplexed Luminex^®^-based immunoassay, as previously described [143], following the manufacturer’s instructions, and analyzed in the BioPlex 200 instrument (BioRad). The table reports the cytokine amounts (estimated from a standard curve using a fifth-order polynomial equation and adjusted for the dilution factor), normalized to the total protein content (determined by a Lowry assay kit (Sigma-Aldrich)) in the corresponding cell lysates and then subtracted for the values obtained in the absence of LPS. Values are expressed as mean ± SEM, *n* = 5 (biological cultures for each genotype). Statistical analysis was based on unpaired two-tailed Student’s *t*-test. * *p*-value < 0.05.

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
