# Peer review of "Microglia in Prion Diseases: Angels or Demons?"

_ijms, 2020, doi:10.3390/ijms21207765_

Round 1

Reviewer 1 Report

The review by Peggion et al., is a quite complete revision of the participation f microglial cells in the course of transmissible spongiform encephalopathy. The manuscript is clearly written, well organized and compiles the most relevant information about microglial involvement in prion diseases. Despite being a complicated topic, due to the contradictory results found in different investigations, I found it clearly explained in the review.

I have only two minor comments or suggestions that could, in my opinion, improve the manuscript. On the one hand, I would include some comment regarding the role of microglia in prion propagation and spreading. This topic has been highly debated in the prion field for decades, and I found there were no references about the  role of microglial cells on allowing prion propagation and spreading of aggregates.

On the other hand and I consider interesting to include some data on the role of microglia in other proteinopathies that cause neurodegeneration and for which prion-like propagation mechanisms have been described. I would like to see a brief description of common or differential roles of microglia in these other pathologies, since common mechanisms have been also proposed in which microglia could play a role.

Author Response

Response to Reviewer 1 Comments

The review by Peggion et al., is a quite complete revision of the participation f microglial cells in the course of transmissible spongiform encephalopathy. The manuscript is clearly written, well organized and compiles the most relevant information about microglial involvement in prion diseases. Despite being a complicated topic, due to the contradictory results found in different investigations, I found it clearly explained in the review.

 I have only two minor comments or suggestions that could, in my opinion, improve the manuscript. On the one hand, I would include some comment regarding the role of microglia in prion propagation and spreading. This topic has been highly debated in the prion field for decades, and I found there were no references about the  role of microglial cells on allowing prion propagation and spreading of aggregates.

 On the other hand and I consider interesting to include some data on the role of microglia in other proteinopathies that cause neurodegeneration and for which prion-like propagation mechanisms have been described. I would like to see a brief description of common or differential roles of microglia in these other pathologies, since common mechanisms have been also proposed in which microglia could play a role.

Response: We thank the Reviewer for the appropriate suggestions. We have now added a comment (and references thereof) to the possible involvement of glial cells in the replication of prions and prion-like proteins involved in other neurodegenerative disorders at the end of the chapter “2.3. Microglial proliferation/activation in prion diseases” (lines 171-185). Although we agree that microglial cells are deeply involved in other neuronal proteinopathies, we decided not to go in much details on such a topic since the IJMS issue is specifically dedicated to prion disorders.

Reviewer 2 Report

The review by Peggion et al. is a rather interesting and comprehensive overview of the role of microglia in prion disease. This review is adequately written and informative. Some of the verbiage could be refined but overall, the text is easy to understand.

The only suggestion I would have for the authors is to include more figures/schematics that summarize some of the more complex information reported in the paper.

Author Response

Response to Reviewer 2 Comments

The review by Peggion et al. is a rather interesting and comprehensive overview of the role of microglia in prion disease. This review is adequately written and informative. Some of the verbiage could be refined but overall, the text is easy to understand.

The only suggestion I would have for the authors is to include more figures/schematics that summarize some of the more complex information reported in the paper.

Response: Following the Reviewer’s suggestion, we now included in the manuscript two additional cartoons (now Figs. 1 & 2) supporting some of the reported information.
